# Serologic Evidence of Circulation of Six Arboviruses (Dengue Virus, Chikungunya Virus, Zika Virus, Rift Valley Virus, Yellow Fever Virus, Crimean-Congo Hemorrhagic Fever Virus) in Four Regions of Burkina Faso, West Africa

**DOI:** 10.3390/tropicalmed10120345

**Published:** 2025-12-09

**Authors:** Armel Moumouni Sanou, Achille Sindimbasba Nikiéma, Aurélie Sausy, Jeoffray Diendéré, Mathuola Nina Genéviève Ouattara, Arielle Bettina Sandra Badiel, Isidore Bonkoungou, Henri Gautier Ouédraogo, Judith M. Hübschen

**Affiliations:** 1Laboratoire de Recherche sur les Maladies Infectieuses et Parasitaires (LR-MIP), Institut de Recherche en Sciences de la Santé (IRSS), Bobo-Dioulasso 01 BP 545, Burkina Faso; 2Clinical and Applied Virology Group, Department of Infection and Immunity, Luxembourg Institute of Health, 4354 Esch-sur-Alzette, Luxembourgjudith.huebschen@lih.lu (J.M.H.); 3Laboratoire de Recherche en Santé Publique et Nutrition (LR-SPN), Institut de Recherche en Sciences de la Santé (IRSS), Bobo-Dioulasso 01 BP 545, Burkina Faso; 4Laboratory of Molecular Biology, Epidemiology and Monitoring of Bacteria and Viruses Transmitted by Food (LaBESTA), University Joseph KI-ZERBO, Ouagadougou 03 BP 7021, Burkina Faso; 5Laboratoire de Recherche sur les Maladies Infectieuses et Parasitaires (LR-MIP), Institut de Recherche en Sciences de la Santé (IRSS), Ouagadougou 03 BP 7192, Burkina Faso

**Keywords:** arbovirus, seroprevalence, risk factors, Burkina Faso

## Abstract

Apart from some information on dengue virus (DENV), there is limited data on the circulation of arboviruses in Burkina Faso. The aim of this study was to investigate antibody prevalence against six arboviruses in four regions of the country to document previous virus exposure. Serum samples collected between August 2018 and December 2022 from people infected with viral hepatitis B and C in Bobo-Dioulasso were used to detect IgG antibodies against DENV, Chikungunya virus (CHIKV), Zika virus (ZIKV), Yellow fever virus (YFV), Rift Valley fever virus (RVFV) and Crimean-Congo hemorrhagic fever virus (CCHFV) using commercial ELISA kits. A total of 1808 serum samples, accompanied by basic epidemiologic data (sex, age and residency) were included in this study. We observed an IgG antibodies seroprevalence of 75.4% for DENV, 30.8% for CHIKV, 2.9% for ZIKV, 1.2% for RVFV, 1.1% for CCHFV and 1.1% for YFV. Age, sex, and place of residence were significantly associated with seropositivity for DENV and age and sex with CHIKV seropositivity. The results suggested widespread circulation of DENV and CHIKV and possible circulation of CCHFV and RVFV in humans in Burkina Faso. The importance of strengthening arbovirus surveillance by including additional arboviruses in the diagnostic panel is emphasized.

## 1. Introduction

Arboviruses are transmitted by arthropod vectors including mosquitoes, ticks, and sandflies [1]. They are responsible for arboviral infections, which are asymptomatic in about 75% of cases. When symptomatic, arboviral infections typically present as an influenza-like syndrome with nonspecific signs and symptoms [2].

Arbovirus infections are more common in tropical countries, where climatic conditions are conducive to the emergence and re-emergence of vectors. However, in recent years, many countries have experienced outbreaks of arboviral diseases due to advanced deforestation, globalization and climate change [3,4]. Currently, arbovirus infections are considered a global public health concern [4,5].

The arboviruses with the greatest impact on humans are dengue (DENV) and Zika viruses (ZIKV) of the family *Flaviviridae* and chikungunya virus (CHIKV) of the family *Togaviridae* [6,7]. DENV, one of the most extensively studied arboviruses, has been estimated to cause between 50 and 100 million cases of dengue fever and 500,000 cases of severe dengue requiring hospitalization each year worldwide [8]. It is present in at least 128 countries and approximately 4 billion people are at risk of infection each year [4,5]. In 2013, the emergence of CHIKV in the Caribbean and its rapid spread to 45 countries and territories in the Americas highlighted the high epidemic potential of this virus [5]. CHIKV is now widespread globally [9]. Despite the low mortality rates associated with CHIKV, infection with this virus has a significant impact on the quality of life and results in significant economic losses [10]. In 2024, as of April 30, approximately 240,000 cases of CHIKV disease and more than 90 deaths have been reported worldwide [11]. First identified in Uganda in 1947, ZIKV has spread rapidly around the world. The 2015 epidemic recorded approximately 1,500,000 suspected cases in the Americas, Caribbean and Pacific [12]. ZIKV infection is usually mild. However, it can cause neurological disorders such as congenital microcephaly, Guillain-Barré syndrome, myelitis, and meningoencephalitis [13]. As for yellow fever (YF), despite the existence of an effective vaccine, it is considered a re-emerging arboviral disease. Indeed, in recent years, YF cases have been continuously reported in tropical countries, especially in Africa, where 16 YF outbreaks have been reported in 13 countries [14]. Clinically, the majority of human infections with yellow fever virus (YFV) are asymptomatic or mild. However, severe YF is estimated to occur in 12% of infected individuals, with a case fatality rate of approximately 47% [15,16]. Rift Valley Fever Virus (RVFV) was discovered in Kenya in 1930 and primarily infects domestic animals (sheep, goats, cattle) and humans [17]. Human infection ranges from self-limiting febrile illness to life-threatening hemorrhagic episodes [18]. Since its discovery, RVFV has caused numerous epidemics in Africa and the Arabian Peninsula [17,19,20]. Crimean-Congo hemorrhagic fever (CCHF) is a zoonosis caused by the Crimean-Congo hemorrhagic fever virus (CCHFV). It belongs to the *Nairoviridae* family and is transmitted to humans mainly by the bite of a tick of the genus *Hyalomma* [21]. Symptomatic patients initially present with nonspecific illness, which can progress to a potentially fatal outcome. Epidemics have been reported, and mortality rates varied from region to region, ranging from about 5% to 80% [22,23].

Although vector-borne diseases pose a serious threat to the health of the world’s population [24], epidemiologic data are limited in Africa, due to limited surveillance, a nonexistent or poor screening system, and low awareness, despite the historical presence and recent epidemics on the continent [25].

Burkina Faso, a tropical country in West Africa, has faced recurrent dengue epidemics since 2013 [26,27,28]. ZIKV and CHIKV circulation in Burkina Faso has recently been demonstrated [29,30] and YFV infections were reported between 2003 and 2008 through the national febrile jaundice surveillance system [31]. To our knowledge, RVFV and CCHFV circulation in the human population has not yet been confirmed in the country. Although Burkina Faso has a surveillance system for arboviruses, this surveillance is mainly focused on DENV, resulting in a knowledge gap regarding other arboviruses. In addition, the few existing data on the circulation of these viruses are old, geographically limited, and involve a small population. We conducted a seroprevalence study for antibodies against DENV, CHIKV, ZIKV, YFV, CCHFV and RVFV in four regions of Burkina Faso to gain an overview of the burden of these infections as a basis for prevention and control strategies.

## 2. Materials and Methods

### 2.1. Study Design

This was a cross-sectional study based on serum samples collected at the Assaut-Hépatites Center, as part of follow-up of patients infected with hepatitis B or C viruses in Bobo-Dioulasso, Burkina Faso. Assaut-Hépatites is a nongovernmental organization (NGO) that operates a referral center for the treatment of viral hepatitis, receiving infected patients from various regions of Burkina Faso daily. Approximately 8 mL of whole blood was collected from each patient by venipuncture. The blood samples were centrifuged at 4000 rpm for five minutes at 25 °C and the resulting serum was aliquoted into two screw-cap tubes. One aliquot was used for molecular analysis of hepatitis viruses and the second was stored at −80 °C for subsequent analyses.

### 2.2. Sample Size Calculation

The target sample size was estimated at 954 using the Schwartz formula: n = $z^{2}p\left(\frac{1-p}{d^{2}}\right)$, where

n is the required sample size,

z is the z-score corresponding to the desired confidence level (1.96 for 95% CI),

p is the expected prevalence (66.3% for anti-DENV IgG antibodies [32]) and the desired precision (0.03).

### 2.3. Sample Selection and Data Collection

For this study, 1808 samples were randomly selected using STATA version 17.0 (StataCorp LLC, College Station, TX, USA) software. Selection criteria included sufficient serum volume, proper labeling, and adequate storage at −80 °C. Sociodemographic data, including sex, age, and place of residence, were extracted from the database. The samples included in the study were collected between August 2018 and December 2022 and were from participants residing in the Hauts Bassins, Boucle du Mouhoun, Sud-Ouest and Cascades regions located in the Western part of Burkina Faso (Figure 1).

### 2.4. Serological Detection

Serum samples were tested for the presence of immunoglobulin G (IgG) class antibodies against DENV, CHIKV, ZIKV, YFV, RVFV and CCHFV using commercial ELISA kits (Table 1). Cut-off values were calculated according to the manufacturers’ instructions. Results were interpreted as negative, equivocal, or positive. Equivocal results were retested and in case they were equivocal again, they were considered as negative for further analyses. Positive and negative controls supplied with the commercial kits were tested on each plate to ensure assay validity.

### 2.5. Statistical Analysis

Data were entered using Microsoft Excel and analyzed using STATA version 17.0 (StataCorp LLC, College Station, TX, USA). Categorical variables were expressed as percentages with 95% CIs; comparisons used chi-square or Fisher’s exact test as appropriate. Quantitative variables (age) were expressed as mean ± SD. Multiple logistic regression analyses were performed to assess associations between participant characteristics (age, sex, and region of residence) and seropositivity to each arbovirus. Crude and adjusted odds ratios (cORs, aORs) with 95% CIs were calculated. A *p*-value < 0.05 was considered statistically significant.

## 3. Results

### 3.1. Study Population

Men accounted for 60.7% of the 1808 hepatitis patients (1097/1808, Table 2). The mean age was 35.7 ± 11.5 years (range 3–83), with the 30–39-year age group most represented (697 participants, 38.0%). Most participants resided in the Hauts Bassins region (n = 1244; 68.8%).

### 3.2. Seroprevalence of Antibodies Against Arboviruses

IgG antibodies against at least one arbovirus were detected in 1472 samples (81.4%; 95% CI: 79.5–83.1). The IgG antibodies seroprevalence was 75.4% (1363/1808; 95% CI: 73.3–77.4) for DENV, 30.7% (556/1808; 95% CI: 28.2–32.5) for CHIKV, 2.9% (53/1808; 95% CI: 2.1–3.7) for ZIKV, 1.2% (21/1808; 95% CI: 0.7–1.7) for RVFV, 1.1% (20/1808; 95% CI: 0.7–1.6) for CCHFV and 1.1% (19/1808; 95% CI: 0.6–1.6) for YFV. A significant increase in seroprevalence with age was observed for DENV from 27.3% among <20-year-olds to 90.5% among ≥50-year-olds (*p* < 0.001) and for CHIKV from 11.4% among <20-year-olds to 38.2% among ≥50-year-olds (*p* < 0.001, Table 3). Furthermore, the prevalence of IgG antibodies for DENV was significantly higher in males (77.2%) than in females (72.6%, *p* = 0.029), whereas the reverse was true for CHIKV (males: 28.8% versus females: 33.8%; *p* = 0.026). Significant differences between regions were observed only for anti-DENV IgG, with prevalence rates ranging from 73.6% in the Boucle du Mouhoun region to 85.3% in the Sud-Ouest region (*p* = 0.002, Table 3).

Seropositivity for IgG antibodies against two or more arboviruses was detected in 724 (724/1471; 49.2%) of the positive samples. Double positivity was observed in 665 (665/1471; 45.2%) and triple positivity in 59 samples (59/1471; 4.0%) (Table 4). Anti-DENV IgG/anti-CHIKV IgG (455/1471; 30.9%) and anti-DENV IgG/anti-ZIKV IgG (50/1471; 3.4%) were the most frequently detected combinations. Among triple positives, anti-DENV IgG/anti-CHIKV IgG/anti-ZIKV IgG (21/1471; 1.4%) and anti-CCHFV IgG/anti-RVFV IgG/anti-YFV IgG (16/1471; 1.1%) were the most common (Figure 2).

### 3.3. Associations Between Participant Characteristics and Seropositivity

Multiple logistic regression analysis was performed to assess the association between certain variables and arbovirus seropositivity. Age, sex and place of residence were significantly associated with DENV seropositivity, while age and sex were associated with CHIKV seropositivity (Table 4). Participants aged 20–29 years were five times more likely to be positive for anti-DENV (aOR: 5.4; 95% CI: 3.2–9.2; *p* = 0.001) and nearly three times more likely for anti-CHIKV (aOR: 2.6; 95% CI: 1.3–5.3; *p* = 0.005) compared with those under 20 years. Seropositivity increased among participants ≥50 years, who were 27 times more likely to be DENV seropositive (aOR: 27.3; 95% CI: 14.5–54.0; *p* = 0.001) and approximately five times more likely to be CHIKV seropositive (aOR: 4.8; 95% CI: 2.3–9.8; *p* = 0.001) than participants under 20 years. Male sex was also positively associated with DENV seropositivity (aOR: 1.2; 95% CI: 1.0–1.6; *p* = 0.01) and negatively associated with CHIKV seropositivity (aOR: 0.8; 95% CI: 0.6–1.0; *p* = 0.03). Regarding place of residence, only living in the Sud-Ouest region was significantly associated with DENV seropositivity (aOR: 2.3; 95% CI: 1.3–4.0; *p* = 0.003) (Table 4).

The following factors were found to be associated with seropositivity to DENV and CHIKV in the study population (N = 724). The study examined a range of variables, including age, gender, and geographical location. Both cOR and aOR with 95% CI are shown to assess associations with arbovirus IgG seropositivity.

## 4. Discussion

The results of our study showed seroprevalences ranging from 1.1% for anti-YFV IgG to 75.4% for anti-DENV IgG. The high seroprevalence of anti-DENV IgG confirms the endemic circulation of DENV in Burkina Faso and supports data reported in previous studies [27,28,32,33]. Our study did not differentiate between serotypes, but previous studies have documented the presence of all 4 of them in the country [8,24,34,35]. The seroprevalence of 30.7% for anti-CHIKV IgG, which is similar to that reported by Lim et al. (29.1%) [29] in their 2015 population based study, also demonstrates wide circulation of CHIKV. Since CHIKV was also detected in vectors from Burkina Faso [36], this virus might be considered as possible cause of febrile infections and be included in the diagnostic panel of local health facilities. In contrast to the recently reported anti-ZIKV IgG seroprevalence of 22.6% in blood donors [30], we found a prevalence of only 2.8%, which may be attributable to differences in study populations (blood donors vs. HBV/HCV patients) and analytical methods (ELISA vs. Luminex – Microneutralization). We found a seroprevalence of 1.1% for anti-YFV IgG. Although the test used could not determine whether it was a result of previous contact or vaccination, national surveillance data confirmed the presence of the virus in Burkina Faso [31].

For the first time, our study provides serological indications of possible exposure to CCHFV (1.1%) and RVFV (1.2%) in the human population in Burkina Faso. Although seroprevalence levels were low, these findings suggest that these viruses may be present in the country and underline the relevance of including them in arbovirus surveillance efforts. Previous evidence of CCHFV exposure in sheep in Burkina Faso has been reported [37]. Additionally, studies from neighboring West African countries (Ghana, Senegal, Mauritania, Nigeria) [24,38,39,40] and other African regions (Cameroon, Tunisia) [41,42] support the potential for circulation of these viruses in both human and animal populations.

Multivariate analysis showed that IgG antibodies seroprevalence for DENV increased significantly with age, possibly because of continuous exposure risk and recurrent DENV epidemics since 2013 [35]. This result is in agreement with other studies conducted in the country [8,32], showing the association between age and seropositivity. The same association was found for anti-CHIKV IgG, suggesting endemic circulation of CHIKV in Burkina Faso and potentially also under detection during epidemics [29].

The seroprevalence was higher in males (77.2%) than in females (72.6%) for anti-DENV IgG and inverse for anti-CHIKV IgG (males: 28.8% versus females: 33.8%), which is consistent with other studies showing a higher risk of arbovirus infection in men (CHIKV) [43,44], (DENV) [45] or in women (DENV) [46]. This difference in exposure according to sex is difficult to explain since both viruses have the same vectors, but may be related to the sociocultural characteristics of each country [47]. In Burkina Faso, women spend most of their time in households that are often close to breeding sites of mosquitoes, which would explain at least the higher exposure to CHIKV, while men might be more exposed to sylvatic DENV.

Th IgG antibodies seroprevalence for DENV was very high in all four regions, ranging from 73% in the Boucle du Mouhoun region to 85% in the Sud-Ouest region. The positive association between residence in the Sud-Ouest region and exposure to dengue virus might be related to the high rainfall in this region, which favors the development and maintenance of vectors.

Although seroprevalence studies are important to assess arbovirus disease burden, the correct interpretation of the results poses a challenge especially for flaviviruses, which share several conserved epitopes, potentially leading to cross-reactions [48,49]. However, the magnitude of these cross-reactions varies depending on the type of flavivirus, the type of test used, and the target protein [50] and has been reported to range between 15.4% and 84% for DENV and other flaviviruses (YFV, West Nile Virus and Japanese encephalitis virus) [50,51], while another study reported no cross-reactivity of antibodies produced by yellow fever vaccine and a ZIKV ELISA and only 3.9% for one of three DENV ELISA kits tested [52]. Cross-reactivity of antibodies against CHIKV and DENV or other flaviviruses has been reported to be low (7%) [53], which is important for our study, where most of the double and triple positives involved DENV and CHIKV. Although the ELISA tests used in our study seemed to have high specificity (Table 1), not all cross-reactions relevant for this study were assessed by the manufacturers. Thus, we cannot exclude a certain overestimation of some antibody prevalence rates, but retain the main information, which is the confirmation of the circulation of the studied arboviruses in Burkina. This is especially relevant for RVFV and CCHFV, for which to the best of our knowledge, no cross-reactions with flaviviruses or togaviruses have been reported and for which we found two single positive samples each. In addition to the potential overestimation of seroprevalence rates, the retrospective nature of the study did not allow to include certain relevant variables (e.g., occupation, travel, living with pets, etc.) to assess associations with arbovirus exposure risk.

It should be noted that the study has some limitations. We were unable to perform confirmatory neutralization tests that would have distinguished actual infections from potentially cross-reactive IgG responses, particularly among flaviviruses. In addition, while the ELISA kits used report high diagnostic performance, their accuracy may vary in the West African context where multiple arboviruses co-circulate. A certain degree of cross-reactivity with related nairoviruses or phleboviruses cannot be excluded, since the ELISA test detects binding antibodies rather than functional antibodies. Finally, the use of samples from patients with viral hepatitis and the lack of information on occupation, travel or contact with animals may limit the generalizability of our findings.

## 5. Conclusions

Our results indicate a high seroprevalence of DENV and CHIKV, suggesting widespread circulation of these viruses in Burkina Faso. IgG antibodies against CCHFV and RVFV were detected, suggesting possible human exposure for the first time in the country; however, confirmatory testing using virus neutralization assays is needed. Strengthening arbovirus surveillance, including additional relevant viruses in diagnostic panels and using sensitive and specific diagnostic tools, will provide a clearer understanding of disease burden and inform prevention and control strategies.

## Figures and Tables

**Figure 1 tropicalmed-10-00345-f001:**
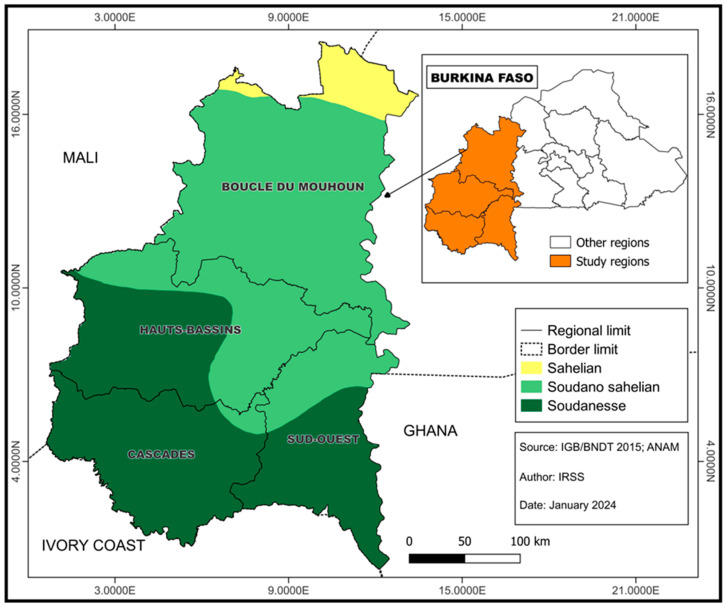
Map showing the study regions.

**Figure 2 tropicalmed-10-00345-f002:**
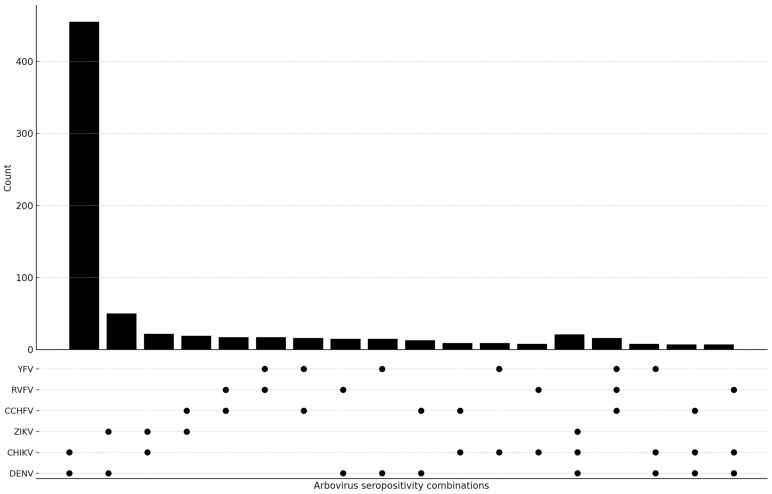
The UpSet plot shows how often dual and triple arbovirus IgG seropositivity patterns appear together, and how much they overlap. The upper bar chart shows the number of participants with each serological combination, while the dot matrix indicates the viruses included in each pattern. The filled circles on the graph show the IgG results. The figure shows how the different viruses (DENV, CHIKV, ZIKV, CCHFV, RVFV and YFV) are exposed to each other.

**Table 1 tropicalmed-10-00345-t001:** Characteristics of the ELISA tests.

	Dengue Virus IgG ELISA	Chikungunya IgG ELISA	Zika Virus IgG (Capture) ELISA	Yellow Fever Virus IgG ELISA	Human Rift Valley Fever Virus IgG ELISA	**Human Crimean-Congo Hemorrhagic Fever Virus** **IgG ELISA**
Manufacturer	DRG Diagnostic, Marburg, Germany	DRG Diagnostic, Marburg, Germany	DRG Diagnostic, Marburg, Germany	DRG Diagnostic, Marburg, Germany	Abbexa LTD, Cambridge, United Kingdom	Abbexa LTD, Cambridge, United Kingdom
Diagnostic Sensitivity	100%	98.7%	96.0%	>98%	-	-
Diagnostic Specificity	100%	100%	99.6%	>98%	-	-
Cross-reactivity	No cross-reactivity in IgG-positive samples for HSV1/2, Rubella, EBV, CMV, VZV, TBE, RSV, Parvovirus	No evidence of false-positive results due to cross-reactivity. Cross-reactivity with antibodies against other alpha viruses cannot be excluded	No evidence of false-positive results due to cross-reactivity	Cross-reactivity has been observed with DENV and ZIKV IgG-positive specimens	-	-

-: No information provided by the manufacturer.

**Table 2 tropicalmed-10-00345-t002:** Sociodemographic characteristics of study participants.

Characteristics	n	%
Sex		
Female	711	39.3
Male	1097	60.7
Age (in years)		
<20	88	4.9
20–29	446	24.7
30–39	687	38.0
40–49	367	20.3
≥50	220	12.2
Residence region		
Hauts Bassins	1244	68.8
Cascades	192	10.6
Sud-Ouest	232	12.8
Boucle du Mouhoun	140	7.7

**Table 3 tropicalmed-10-00345-t003:** Seroprevalence of IgG antibodies against arboviruses by sociodemographic characteristics.

Characteristics	Total	Anti-DENV IgG		Anti-CHIKV IgG		Anti-ZIKV IgG		Anti-CCHFV IgG		Anti-RVFV		Anti-YFV IgG	
		Positive n (%)	p	Positive n (%)	*p*	Positive n (%)	*p*	Positive n (%)	*p*	Positive n (%)	*p*	Positive n (%)	*p*
Seroprevalence	N = 1808	1363 (75.4)		556 (30.7)		53 (2.9)		20 (1.1)		21 (1.2)		19 (1.1)	
Age (in years)			<0.001		<0.001		0.540		0.483		0.185		0.141
<20	88	24 (27.3)		10 (11.4)		0		0		1 (1.1)		0	
20–29	446	294 (65.9)		111 (24.9)		7 (1.6)		4 (0.9)		3 (0.7)		3 (0.7)	
30–39	687	530 (77.1)		229 (33.3)		17 (2.5)		6 (0.9)		5 (0.7)		4 (0.6)	
40– 49	367	316 (86.1)		122 (33.2)		20 (5.4)		6 (1.6)		8 (2.2)		7 (1.9)	
≥50	220	199 (90.5)		84 (38.2)		9 (4.1)		4 (1.8)		4 (1.8)		5 (2.3)	
Sex			0.029		0.026		0.167		0.601		0.434		0.096
Female	711	516 (72.6)		240 (33.8)		16 (2.2)		9 (1.3)		10 (1.4)		11 (1.5)	
Male	1097	847 (77.2)		316 (28.8)		37 (3.4)		11 (1.0)		11 (1.0)		8 (0.7)	
Residence region			0.002		0.055		0.353		0.496		0.092		0.193
Boucle du Mouhoun	140	103 (73.6)		42 (30.0)		2 (1.4)		0		0		0	
Hauts Bassins	1244	919 (73.9)		363 (29.2)		35 (2.8)		14 (1.1)		14 (1.1)		13 (1.0)	
Cascades	192	143 (74.5)		74 (38.5)		9 (4.7)		2 (1.0)		1 (0.5)		1 (0.5)	
Sud-Ouest	232	198 (85.3)		77 (33.2)		7 (3.0)		4 (1.7)		6 (2.6)		5 (2.2)	

**Table 4 tropicalmed-10-00345-t004:** Factors associated with seropositivity against arboviruses.

Characteristics	Anti-DENV IgG				Anti-CHIKV IgG			
	cOR ^#^ (95% CI)	*p*	aOR * (95% CI)	*p*	cOR ^#^ (95% CI)	** *p* **	**aOR * (95% CI)**	** *p* **
Age (in years)								
<20	1	-	1	-	1	-	1	-
20–29	5.1 (3.1–8.6)	0.001	5.4 (3.2–9.2)	0.001	2.5 (1.3–5.4)	0.005	2.6 (1.3–5.3)	0.005
30–39	8.9 (5.4–15.0)	0.001	9.4 (5.6–15.6)	0.001	3.8 (2.0–8.0)	0.001	3.9 (1.9–7.7)	0.001
40–49	16.3 (9.4–28.9)	0.001	16.9 (9.8–30.2)	0.001	3.8 (1.9–8.1)	0.001	3.8 (1.9–7.7)	0.001
≥50	24.6 (13.1–48.5)	0.001	27.3 (14.5–54.0)	0.001	4.7 (2.4–10.2)	0.001	4.8 (2.3–9.8)	0.001
Sex								
Female	1		1		1		1	
Male	1.3 (1.0–1.6)	0.02	1.2 (1.0–1.6)	0.01	0.8 (0.6–1.0)	0.02	0.8 (0.6–1.0)	0.030
Residence region								
Boucle du Mouhoun	1		1		1		1	
Cascades	1.0 (0.6–1.7)	0.85	1.0 (0.6–1.6)	0.97	1.5 (0.9–2.3)	0.13	1.4 (0.9–2.3)	0.13
Hauts-Bassins	1.0 (0.7–1.5)	0.95	1.1 (0.7–1.6)	0.89	1.0 (0.6–1.4)	0.79	1.0 (0.6–1.4)	0.83
Sud-Ouest	2.1 (1.2–3.5)	0.006	2.3 (1.3–4.0)	0.003	1.1 (0.7–1.8)	0.54	1.2 (0.7–1.8)	0.51

^#^ crude odds ratio, * adjusted odds ratio.

## Data Availability

The authors confirm that the relevant data generated during this study are included in the article. All data supporting these results are available from the corresponding author upon reasonable request.

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
