# Peer review of "Serologic Evidence of Circulation of Six Arboviruses (Dengue Virus, Chikungunya Virus, Zika Virus, Rift Valley Virus, Yellow Fever Virus, Crimean-Congo Hemorrhagic Fever Virus) in Four Regions of Burkina Faso, West Africa"

_tropicalmed, 2025, doi:10.3390/tropicalmed10120345_

Round 1
Reviewer 1 Report
Comments and Suggestions for Authors
Authors have addressed a critical component of arbovirus epidemiology and detection. The paper is well-written with clear assumptions throughout the manuscript. Aspects addressed are critical to provide an almost comprehensive a background of arboviruses circulation in Burkina Faso.
Minor
- Use the adequate checklist for this study.
- Arrange the methods section in a logical pathway.
- What was the algorithm whenever a repeat was equivocal?
- Lines 114-116: samples were randomly selected using STATA software…Line 143: Variables were entered using Excel and analyzed using STATA 17.0 (Texas, USA). These sentences look contradictory. Correct.
- Lines 253-255: “In Burkina Faso, women spend most of their time in households that are often close to breeding sites of mosquitoes, which would explain at least the higher exposure to CHIKV, while men might be more exposed to sylvatic DENV”. If this is the hypothesis, it means that we should also have more men infected with Chik due to exposure to breeding sites of mosquitoes around houses. If this is not the case, kindly rephrase this sentence.
- Add a prospective on diagnostic tools to be used in the detection of arboviruses and the determination of their real burden in the region/country. Then, authors can suggest not only the integration of new viruses to the testing panel, but overall, they can really bring a relevant contribution to the detection of circulation viruses in the country.
- Correct references 3, 4, 8, 12, etc.

Some phrases need to be improved but in general the paper is well-written.
Author Response
|
Response to Reviewer 1 Comments |
||
|
1. Summary |
|
|
|
Thank you very much for taking the time to review this manuscript. Please find the detailed responses below and the corresponding revisions/corrections highlighted/in track changes in the re-submitted files |
||
|
2. Point-by-point response to Comments and Suggestions for Authors
|
||
|
Comments 1: General Owerview Authors have addressed a critical component of arbovirus epidemiology and detection. The paper is well-written with clear assumptions throughout the manuscript. Aspects addressed are critical to provide an almost comprehensive a background of arboviruses circulation in Burkina Faso. |
||
|
Response 1: The authors are grateful to the reviewer for their encouraging comments. |
||
|
Comments 2: Use the adequate checklist for this study.
|
||
|
Response 2: The adequate checklist has been used for this study
Comments 3: Arrange the methods section in a logical pathway. Response 3: We thank the reviewer for this pertinent comment. The ‘Materials and Methods’ section has been organized as follows: Study design – Sample size calculation – Samples selection and data collection – serological detection
Comments 4: What was the algorithm whenever a repeat was equivocal? Response 4: We thank the reviewer for their question. Whenever the repetition was equivocal, we considered it negative.
Comments 5: Lines 114-116: samples were randomly selected using STATA software…Line 143: Variables were entered using Excel and analyzed using STATA 17.0 (Texas, USA). These sentences look contradictory. Correct. Response 5: We thank the reviewer for this clarification request. STATA software was initially used for random sample selection, and subsequently for statistical analysis.
Comments 6: Lines 253-255: “In Burkina Faso, women spend most of their time in households that are often close to breeding sites of mosquitoes, which would explain at least the higher exposure to CHIKV, while men might be more exposed to sylvatic DENV”. If this is the hypothesis, it means that we should also have more men infected with Chik due to exposure to breeding sites of mosquitoes around houses. If this is not the case, kindly rephrase this sentence.
Response 6: The authors thank the reviewer for their comment and suggestion. However, this hypothesis does not apply to men in the Burkinabè context. The majority of women are housewives and, as a consequence, they spend more time in proximity to mosquito breeding sites. Conversely, men typically depart in the morning to engage in their daily activities, returning home at night.
Comments 7: Add a prospective on diagnostic tools to be used in the detection of arboviruses and the determination of their real burden in the region/country. Then, authors can suggest not only the integration of new viruses to the testing panel, but overall, they can really bring a relevant contribution to the detection of circulation viruses in the country. Response 7: We thank the reviewer for this pertinent observation. A sentence has been added to the conclusion section suggesting the use of sensitive and specific diagnostic tools in the surveillance system.
Comments 8: Correct references 3, 4, 8, 12, etc. Response 8: All references have been corrected as requested.
|
||

Reviewer 2 Report
Comments and Suggestions for Authors
General Overview
Sanou et al. conducted a cross-sectional seroepidemiological study to assess the circulation of six medically important arboviruses in Burkina Faso, where data, particularly beyond dengue virus (DENV), remain scarce. The study analyzed 1,808 serum samples collected between August 2018 and December 2022 from individuals infected with hepatitis B or C in Bobo-Dioulasso. Using commercial ELISA kits, the authors screened for IgG antibodies against DENV, Chikungunya virus (CHIKV), Zika virus (ZIKV), Yellow fever virus (YFV), Rift Valley fever virus (RVFV), and Crimean-Congo hemorrhagic fever virus (CCHFV).
The results revealed high seroprevalence of DENV (75.4%) and moderate prevalence of CHIKV (30.8%), while antibodies to ZIKV (2.9%), RVFV (1.2%), CCHFV (1.1%), and YFV (1.1%) were detected at low levels. Statistical analyses identified significant associations between age, sex, and residence with DENV seropositivity, and age and sex with CHIKV seropositivity.
The study provides valuable serological data on arbovirus exposure in Burkina Faso. However, the claim of “first evidence” of human exposure to CCHFV and RVFV should be interpreted cautiously, as detection was based solely on ELISA IgG assays, which can be prone to cross-reactivity. Confirmation with gold-standard assays, such as virus neutralization tests, is necessary to definitively establish circulation.
Below are my detailed suggestions organized by manuscript sections:
Title
Lines 1–4: Consider a more concise and appealing title, e.g., Serological evidence of arbovirus circulation in Burkina Faso, West Africa.
Abstract
Lines 33–34: Clarify phrasing: “A total of 1,808 serum samples, accompanied by basic epidemiological data (sex, age, and residency), were included in the study.”
Line 34: Add “We observed a seroprevalence of…” to improve clarity.
Line 37: Correct “seropositivity for anti-DENV” to “seropositivity for DENV” and check the text for similar instances.
Line 38: Replace “wide circulation” with “widespread circulation.”
Lines 38–39: Reconsider the statement on first evidence of CCHFV and RVFV; see discussion/conclusion comments.
Introduction
The introduction is comprehensive but could be more concise. Some sentences are dense with numbers and historical details; certain points, such as the widespread nature of DENV or CHIKV’s impact on quality of life, are repeated and could be condensed. The main research question (seroprevalence of six arboviruses) should appear earlier. Suggested minor edits:
Lines 44–47: Replace “arboviral diseases” with “arboviral infections” and “initially present” with “typically present.”
Lines 48–53: “Emergence of arboviral diseases” → “outbreaks of arboviral diseases”; “scourge” → “public health concern.”
Lines 54–64: Break long sentences; “has now spread throughout the world” → “is now widespread globally.”
Lines 65–70: “Benign in most cases” → “usually mild.”
Lines 81–85: Clarify: “Symptomatic patients initially present with nonspecific illness, which can progress to a potentially fatal outcome.”
Materials and Methods
I suggest adding a subsection describing the sample size calculation to demonstrate that the number of samples included is adequate to estimate the seroprevalence of each arbovirus with sufficient precision. Providing this information will strengthen the methodological rigor of the study and allow readers to assess whether the sample is representative of the target population.
Below the standard formula used for estimating sample size in prevalence studies:
n=Z2×p×(1−p)/d2
where:
n is the required sample size,
Z is the Z-score corresponding to the desired confidence level (typically 1.96 for 95% CI),
p is the expected prevalence (based on previous studies or a pilot estimate),
d is the desired precision (margin of error), often set between 0.02 and 0.05.
Including this information in the Methods section would make it clear that the number of samples analyzed is sufficient to reliably estimate arbovirus seroprevalence in the study population.
Lines 104–106: Specify “serum samples” and replace “people” with “patients.” Suggested: This was a cross-sectional study based on serum samples collected at the Assaut-Hépatites Center, as part of follow-up of patients infected with hepatitis B or C viruses in Bobo-Dioulasso, Burkina Faso.
Lines 107–108: Suggested: Assaut-Hépatites is a non-governmental organization (NGO) that operates a referral center for the treatment of viral hepatitis, receiving infected patients from various regions of Burkina Faso daily.
Lines 109–111: Add a comma in “4,000 rpm” and specify centrifugation temperature. Use “screw-cap tubes” instead of “thread tubes.”
Lines 112–113: “Subsequent analyses” reads better than “further analysis.”
Lines 114–116: Clarify selection criteria: For this study, 1,808 samples were randomly selected using STATA software. Selection criteria included sufficient serum volume, proper labeling, and adequate storage at −80°C. Sociodemographic data, including sex, age, and place of residence, were extracted from the database.
Line 119: Add “located in the western part” for clarity.
Line 120: If the figure is original, indicate software used; if sourced, check permissions. Include sample proportions per region and Burkina Faso’s position in Africa.
Lines 126–130: Add “(IgG)” at first mention of immunoglobulin G; rephrase controls as: Positive and negative controls supplied with the commercial kits were tested on each plate to ensure assay validity.
Line 136: Table 1 could include more details on antigens and references of each kit.
Statistical Analysis
Lines 142–149: Suggested re-phrasing:
Data were entered in Microsoft Excel and analyzed using Stata version 17.0 (StataCorp LLC, College Station, TX, USA). Categorical variables were expressed as percentages with 95% CIs; comparisons used chi-square or Fisher’s exact test as appropriate. Quantitative variables (age) were expressed as mean ± SD. Multiple logistic regression analyses were performed to assess associations between participant characteristics (age, sex, and region of residence) and seropositivity to each arbovirus. Crude and adjusted odds ratios (cORs, aORs) with 95% CIs were calculated. A p-value < 0.05 was considered statistically significant.
Results
Lines 153–157: Suggested rewording: Men accounted for 60.7% of the 1,808 hepatitis patients (1,097/1,808, Table 2). The mean age was 35.7 ± 11.5 years (range 3–83), with the 30–39-year age group most represented (697 participants, 38.0%). Most participants resided in the Hauts Bassins region (n = 1,244; 68.8%).
Line 161: Clarify IgG seroprevalence: IgG antibodies against at least one arbovirus were detected in 1,472 samples (81.4%; 95% CI: 79.5–83.1).
Lines 162–181: Minor edits for clarity and consistency; ensure seroprevalence refers to IgG.
Line 183: Tables should include percentages, statistical tests used, and clear titles. Check YFV seroprevalence (likely 19 samples, not 10).
Line 189: Consider replacing Table 4 with an UpSet plot for clarity. Or find a better way to represent data from Table 4.
Lines 195–208: Use “assess the association” instead of “measure”; combine repetitive sentences; correct phrasing and p-values; maintain consistent epidemiological terminology.
Line 211: Legends should specify which factors and viruses are included and the number of participants analyzed.
Discussion
Overall, I recommend revising the discussion with particular care to avoid overinterpretation of the results. Statements should remain cautious, especially when conclusions rely solely on IgG ELISA data.
Lines 122–124: This statement is unnecessary and does not add meaningful content to the discussion. I suggest removing it.
Line 225: When referencing the Lim et al. study (29.1% anti-CHIKV IgG seroprevalence), please specify the study population and year of sampling to allow a more appropriate comparison with your results.
Line 226: Replace “should be considered” with “might be considered” or “could be considered” to avoid overinterpreting the epidemiological implications of your findings.
Line 230: Clarify what the “differences in study population and analytical methods” refer to. Specify how your population differs from that in the cited study and indicate whether different ELISA kits, antigens, or cut-off values were used.
Lines 230–232: The discussion should address possible explanations for the low YFV seroprevalence you observed. For example, consider whether the ELISA used may have low sensitivity, or discuss local vaccination coverage, which can strongly influence anti-YFV IgG prevalence.
Lines 234–237: The statements regarding CCHFV and RVFV need to be significantly softened. Since results are based only on IgG ELISA, cross-reactivity cannot be completely ruled out, and no confirmatory neutralization tests were performed. I recommend replacing the current text with a more cautious formulation, such as:
“For the first time, our study provides serological indications of possible exposure to CCHFV (1.1%) and RVFV (1.2%) in the human population in Burkina Faso. Although seroprevalence levels were low, these findings suggest that these viruses may be present in the country and underline the relevance of including them in arbovirus surveillance efforts. Previous evidence of CCHFV exposure in sheep in Burkina Faso has been reported [37]. Additionally, studies from neighboring West African countries (Ghana, Senegal, Mauritania, Nigeria) [24], [38]–[40] and other African regions (Cameroon, Tunisia) [41], [42] support the potential for circulation of these viruses in both human and animal populations.”
Lines 248–251: When discussing sex-specific differences in seroprevalence, clearly indicate which studies showed higher risk in men and which in women. It would be important to address each virus individually, as your data show divergent trends for DENV and CHIKV. This contextualization will help readers better understand your findings.
Lines 269–281: I recommend adding a dedicated paragraph clearly describing the limitations of your study. Although you mention potential cross-reactivity, this should be explicitly presented as a limitation. In particular, highlight that no confirmatory neutralization assays (e.g., PRNT) could be performed, which restricts the ability to differentiate true past infection from cross-reactive antibody responses.
It would also be useful to discuss the performance of the ELISA kits used in comparison with other studies. Manufacturer-reported sensitivity and specificity do not always reflect real-world performance and may vary depending on population characteristics or co-circulating viruses.
Finally, although cross-reactivity for RVFV and CCHFV with flaviviruses and alphaviruses is generally considered low, antibodies may still cross-react with related nairoviruses or phleboviruses. Since ELISA detects binding IgG without functional confirmation, results should be interpreted as suggestive rather than definitive evidence of prior infection.
Conclusion
The claim of “first evidence” of human CCHFV and RVFV exposure should be softened. ELISA IgG alone may produce cross-reactivity and does not definitively confirm circulation. Recommend adding a statement like:
Our results indicate a high seroprevalence of DENV and CHIKV, suggesting widespread circulation of these viruses in Burkina Faso. IgG antibodies against CCHFV and RVFV were detected, suggesting possible human exposure for the first time in the country; however, confirmatory testing using virus neutralization assays is needed. Strengthening arbovirus surveillance, including additional relevant viruses in diagnostic panels, will provide a clearer understanding of disease burden and inform prevention and control strategies.
Dear authors, I have made several suggestions to enhance the clarity and quality of the English language in your manuscript. I hope you find these recommendations useful.
Author Response
Response to Reviewer 2 Comments
|
1. Summary |
|
|
|
Thank you very much for taking the time to review this manuscript. Please find the detailed responses below and the corresponding revisions/corrections highlighted/in track changes in the re-submitted files.
|
||
|
2. Point-by-point response to Comments and Suggestions for Authors
|
||
|
Comments 1: General Overview
Sanou et al. conducted a cross-sectional seroepidemiological study to assess the circulation of six medically important arboviruses in Burkina Faso, where data, particularly beyond dengue virus (DENV), remain scarce. The study analyzed 1,808 serum samples collected between August 2018 and December 2022 from individuals infected with hepatitis B or C in Bobo-Dioulasso. Using commercial ELISA kits, the authors screened for IgG antibodies against DENV, Chikungunya virus (CHIKV), Zika virus (ZIKV), Yellow fever virus (YFV), Rift Valley fever virus (RVFV), and Crimean-Congo hemorrhagic fever virus (CCHFV). The results revealed high seroprevalence of DENV (75.4%) and moderate prevalence of CHIKV (30.8%), while antibodies to ZIKV (2.9%), RVFV (1.2%), CCHFV (1.1%), and YFV (1.1%) were detected at low levels. Statistical analyses identified significant associations between age, sex, and residence with DENV seropositivity, and age and sex with CHIKV seropositivity. The study provides valuable serological data on arbovirus exposure in Burkina Faso. However, the claim of “first evidence” of human exposure to CCHFV and RVFV should be interpreted cautiously, as detection was based solely on ELISA IgG assays, which can be prone to cross-reactivity. Confirmation with gold-standard assays, such as virus neutralization tests, is necessary to definitively establish circulation. We thank the reviewer for this pertinent comment. We have therefore replaced ‘evidence’ with ‘possible’.
|
||
|
Response 1: The authors would like to express their gratitude to the reviewer 2 for their constructive comments, which will contribute to enhancing the document's quality. |
||
|
Comments 2: Title
Lines 1–4: Consider a more concise and appealing title, e.g., Serological evidence of arbovirus circulation in Burkina Faso, West Africa.
|
||
|
Response 2: We thank the reviewer for their suggestion. The title was proposed in order to be more precise and informative for the reader. We suggest that the proposed title could be the running title of the article.
Comments 3: Abstract
Lines 33–34: Clarify phrasing: “A total of 1,808 serum samples, accompanied by basic epidemiological data (sex, age, and residency), were included in the study.” It is done.
Line 34: Add “We observed a seroprevalence of…” to improve clarity. It is done.
Line 37: Correct “seropositivity for anti-DENV” to “seropositivity for DENV” and check the text for similar instances. It is done.
Line 38: Replace “wide circulation” with “widespread circulation.” It is done.
Lines 38–39: Reconsider the statement on first evidence of CCHFV and RVFV; see discussion/conclusion comments. It is done.
Response 3: We thank the reviewer for these pertinent suggestions, which have been taken into account in the revised version of the article.
Comments 4: Introduction
The introduction is comprehensive but could be more concise. Some sentences are dense with numbers and historical details; certain points, such as the widespread nature of DENV or CHIKV’s impact on quality of life, are repeated and could be condensed. The main research question (seroprevalence of six arboviruses) should appear earlier. It is done.
Suggested minor edits:
Lines 44–47: Replace “arboviral diseases” with “arboviral infections” and “initially present” with “typically present.” It is done.
Lines 48–53: “Emergence of arboviral diseases” → “outbreaks of arboviral diseases”; “scourge” → “public health concern.” It is done.
Lines 54–64: Break long sentences; “has now spread throughout the world” → “is now widespread globally.” It is done.
Lines 65–70: “Benign in most cases” → “usually mild.” It is done.
Lines 81–85: Clarify: “Symptomatic patients initially present with nonspecific illness, which can progress to a potentially fatal outcome.” It is done.
Response 4: We thank the reviewer for these pertinent comments in the introduction section. All comments have been considered in the revised version of the article.
Comments 5: Materials and Methods
I suggest adding a subsection describing the sample size calculation to demonstrate that the number of samples included is adequate to estimate the seroprevalence of each arbovirus with sufficient precision. Providing this information will strengthen the methodological rigor of the study and allow readers to assess whether the sample is representative of the target population. Below the standard formula used for estimating sample size in prevalence studies: n=Z2×p×(1−p)/d2 where: n is the required sample size, Z is the Z-score corresponding to the desired confidence level (typically 1.96 for 95% CI), p is the expected prevalence (based on previous studies or a pilot estimate), d is the desired precision (margin of error), often set between 0.02 and 0.05.
Including this information in the Methods section would make it clear that the number of samples analyzed is sufficient to reliably estimate arbovirus seroprevalence in the study population.
A subsection describing the sample size calculation has been added.
Lines 104–106: Specify “serum samples” and replace “people” with “patients.” Suggested: This was a cross-sectional study based on serum samples collected at the Assaut-Hépatites Center, as part of follow-up of patients infected with hepatitis B or C viruses in Bobo-Dioulasso, Burkina Faso. It is done.
Lines 107–108: Suggested: Assaut-Hépatites is a non-governmental organization (NGO) that operates a referral center for the treatment of viral hepatitis, receiving infected patients from various regions of Burkina Faso daily. It is done.
Lines 109–111: Add a comma in “4,000 rpm” and specify centrifugation temperature. Use “screw-cap tubes” instead of “thread tubes.” It is done.
Lines 112–113: “Subsequent analyses” reads better than “further analysis.” It is done.
Lines 114–116: Clarify selection criteria: For this study, 1,808 samples were randomly selected using STATA software. Selection criteria included sufficient serum volume, proper labeling, and adequate storage at −80°C. Sociodemographic data, including sex, age, and place of residence, were extracted from the database. It is done.
Line 119: Add “located in the western part” for clarity. It is done.
Line 120: If the figure is original, indicate software used; if sourced, check permissions. Include sample proportions per region and Burkina Faso’s position in Africa. We would like to express our gratitude to the reviewer for the suggestions made. The figure is original and was generated using standard graphical functions. With regard to the incorporation of regional sample proportions (sample numbers per region are already provided in table 2) and Burkina Faso's geographic position, these elements are not essential to the figure's purpose and would introduce unnecessary complexity. Consequently, we advocate maintaining the status quo.
Lines 126–130: Add “(IgG)” at first mention of immunoglobulin G; rephrase controls as: Positive and negative controls supplied with the commercial kits were tested on each plate to ensure assay validity. It is done.
Line 136: Table 1 could include more details on antigens and references of each kit. Information on antigens is not provided with the kits.
Response 5: We thank the reviewer for these pertinent comments in the Materials and Methods section. All comments have been considered in the revised version of the article.
Comments 6: Statistical analyses
Lines 142–149: Suggested re-phrasing:
Response 6: We thank the reviewer for this pertinent suggestion. The section has been re-phrased as suggested.
Comments 7: Results
Lines 153–157: Suggested rewording: Men accounted for 60.7% of the 1,808 hepatitis patients (1,097/1,808, Table 2). The mean age was 35.7 ± 11.5 years (range 3–83), with the 30–39-year age group most represented (697 participants, 38.0%). Most participants resided in the Hauts Bassins region (n = 1,244; 68.8%). It is done.
Line 161: Clarify IgG seroprevalence: IgG antibodies against at least one arbovirus were detected in 1,472 samples (81.4%; 95% CI: 79.5–83.1). It is done.
Lines 162–181: Minor edits for clarity and consistency; ensure seroprevalence refers to IgG. It is done.
Line 183: Tables should include percentages, statistical tests used, and clear titles. Check YFV seroprevalence (likely 19 samples, not 10). It is done.
Line 189: Consider replacing Table 4 with an UpSet plot for clarity. Or find a better way to represent data from Table 4. It is done.
Lines 195–208: Use “assess the association” instead of “measure”; combine repetitive sentences; correct phrasing and p-values; maintain consistent epidemiological terminology. It is done. It is done.
Line 211: Legends should specify which factors and viruses are included and the number of participants analyzed. It is done.
Response 7: We thank the reviewer for these pertinent comments in the results section. All comments have been considered in the revised version of the article.
Comments 8: Discussion
Overall, I recommend revising the discussion with particular care to avoid overinterpretation of the results. Statements should remain cautious, especially when conclusions rely solely on IgG ELISA data. It is done.
Lines 222–224: This statement is unnecessary and does not add meaningful content to the discussion. I suggest removing it.
As suggested, this statement has been removed.
Line 225: When referencing the Lim et al. study (29.1% anti-CHIKV IgG seroprevalence), please specify the study population and year of sampling to allow a more appropriate comparison with your results.
Requested information have been added.
Line 226: Replace “should be considered” with “might be considered” or “could be considered” to avoid overinterpreting the epidemiological implications of your findings.
It is done.
Line 230: Clarify what the “differences in study population and analytical methods” refer to. Specify how your population differs from that in the cited study and indicate whether different ELISA kits, antigens, or cut-off values were used.
Precision on study populations and tests used has been inserted in the text.
Lines 230–232: The discussion should address possible explanations for the low YFV seroprevalence you observed. For example, consider whether the ELISA used may have low sensitivity, or discuss local vaccination coverage, which can strongly influence anti-YFV IgG prevalence. We thank the reviewer for the comment. According to the manufacturer, the assay has a sensitivity of more than 98% and thus the low seroprevalence might be due to low vaccination coverage and low exposure in the study population
Lines 234–237: The statements regarding CCHFV and RVFV need to be significantly softened. Since results are based only on IgG ELISA, cross-reactivity cannot be completely ruled out, and no confirmatory neutralization tests were performed. I recommend replacing the current text with a more cautious formulation, such as: “For the first time, our study provides serological indications of possible exposure to CCHFV (1.1%) and RVFV (1.2%) in the human population in Burkina Faso. Although seroprevalence levels were low, these findings suggest that these viruses may be present in the country and underline the relevance of including them in arbovirus surveillance efforts. Previous evidence of CCHFV exposure in sheep in Burkina Faso has been reported [37]. Additionally, studies from neighboring West African countries (Ghana, Senegal, Mauritania, Nigeria) [24], [38]–[40] and other African regions (Cameroon, Tunisia) [41], [42] support the potential for circulation of these viruses in both human and animal populations.”
It is done
Lines 248–251: When discussing sex-specific differences in seroprevalence, clearly indicate which studies showed higher risk in men and which in women. It would be important to address each virus individually, as your data show divergent trends for DENV and CHIKV. This contextualization will help readers better understand your findings.
It is done
Lines 269–281: I recommend adding a dedicated paragraph clearly describing the limitations of your study. Although you mention potential cross-reactivity, this should be explicitly presented as a limitation. In particular, highlight that no confirmatory neutralization assays (e.g., PRNT) could be performed, which restricts the ability to differentiate true past infection from cross-reactive antibody responses.
It would also be useful to discuss the performance of the ELISA kits used in comparison with other studies. Manufacturer-reported sensitivity and specificity do not always reflect real-world performance and may vary depending on population characteristics or co-circulating viruses.
Finally, although cross-reactivity for RVFV and CCHFV with flaviviruses and alphaviruses is generally considered low, antibodies may still cross-react with related nairoviruses or phleboviruses. Since ELISA detects binding IgG without functional confirmation, results should be interpreted as suggestive rather than definitive evidence of prior infection.
We thank the reviewer for these constructive suggestions. A paragraph describing the limitations of your study has been added
Response 8: We thank the reviewer for this pertinent suggestion. The section has been re-phrasing as suggested.
Comment 9: Conclusion
The claim of “first evidence” of human CCHFV and RVFV exposure should be softened. ELISA IgG alone may produce cross-reactivity and does not definitively confirm circulation. Recommend adding a statement like:
Response 9: The authors would like to express their gratitude to the reviewer for this pertinent comment. The conclusion has been amended as suggested.
|
||

Round 2
Reviewer 2 Report
Comments and Suggestions for Authors
The authors have successfully addressed all the suggestions and have responded appropriately to the questions. I recommend accepting the article in its current form.